# Trends and seasonality of 2019–2023 global methane emissions inferred from a localized ensemble transform Kalman filter (CHEEREIO v1.3.1) applied to TROPOMI satellite observations

Drew C. Pendergrass<sup>1</sup>, Daniel J. Jacob<sup>1</sup>, Nicholas Balasus<sup>1</sup>, Lucas Estrada<sup>1</sup>, Daniel J. Varon<sup>1</sup>, James D. East<sup>1</sup>, Megan He<sup>1</sup>, Todd A. Mooring<sup>2</sup>, Elise Penn<sup>2</sup>, Hannah Nesser<sup>3</sup>, and John R. Worden<sup>3</sup>

Correspondence to: Drew Pendergrass (pendergrass@g.harvard.edu)

**Abstract.** We use 2019-2023 TROPOMI satellite observations of atmospheric methane to quantify global methane emissions at monthly 2°×2.5° resolution with a localized ensemble transform Kalman filter (LETKF) inversion, deriving monthly posterior estimates of emissions and year-to-year evolution. We apply two alternative wetland inventories (WetCHARTs and LPJ-wsl) as prior estimates. Our best posterior estimate of global emissions shows a surge from 560 Tg a<sup>-1</sup> in 2019 to 587-592 Tg a<sup>-1</sup> in 2020-2021 before declining to 572-570 Tg a<sup>-1</sup> in 2022-2023. Posterior emissions reproduce the observed 2019-2023 trends in methane concentrations at NOAA surface sites and from TROPOMI with minimal regional bias. Consistent with previous studies, we attribute the 2020-2021 methane surge to a 14 Tg a<sup>-1</sup> increase in emissions from sub-Saharan Africa but find that previous attribution of this surge to anthropogenic sources (livestock) reflects errors in the assumed wetland spatial distribution. Correlation with GRACE-FO inundation data suggests that wetlands in South Sudan played a major role in the 2020-2021 surge but are poorly represented in wetland models. By contrast, boreal wetland emissions decreased over 2020-2023 consistent with drying measured by GRACE-FO. We find that the global seasonality of methane emissions is driven by northern tropical wetlands and peaks in September, later than the July wetland model peak and consistent with GRACE-FO. We find no global seasonality in oil/gas emissions, but US fields show elevated cold season emissions that could reflect increased leakage.

Plain language summary. We use satellite observations of atmospheric methane, a potent greenhouse gas, to calculate emissions from both human and natural sources. We find that methane emissions surged in 2020 and 2021 before declining in 2022 and 2023. We attribute the surge in large part to emissions from eastern Africa, which experienced large methane-generating floods. Wetland models greatly underestimate emissions in that region, which has led some previous work to incorrectly attribute the African surge in methane emissions to livestock.

<sup>&</sup>lt;sup>1</sup> School of Engineering and Applied Sciences, Harvard University, Cambridge, MA, USA

<sup>&</sup>lt;sup>2</sup> Department of Earth and Planetary Sciences, Harvard University, Cambridge, MA, USA

<sup>&</sup>lt;sup>3</sup> Jet Propulsion Laboratory, California Institute of Technology, Pasadena, CA, USA

# 1 Introduction

Methane is a strong greenhouse gas, contributing 0.6 °C of warming from the pre-industrial baseline, with a relatively short lifetime of about 9 years due principally to oxidation by the hydroxyl (OH) radical in the troposphere (Prather et al., 2012; Naik et al., 2021). Methane is emitted by natural sources, mostly wetlands, and by anthropogenic sources including enteric fermentation and manure from livestock, oil and gas, coal mining, rice, landfills, and wastewater (Saunois et al., 2025). Decreasing methane emissions is an effective way to mitigate climate change in the near-term while also achieving air quality co-benefits from reduced tropospheric ozone (West et al., 2006; Nisbet et al., 2020). Bottom-up methane emission inventories link emissions to processes (IPCC, 2019), but inventory construction typically lags by several years behind real time and is subject to errors. Satellite observations of atmospheric methane can help improve and update inventories through inverse analyses using Bayesian optimization and can offer insights on recent and rapid changes (Jacob et al., 2016; Houweling et al., 2017; Jacob et al., 2022). Here we apply a Localized Ensemble Transform Kalman Filter (LETKF) to TROPOspheric Monitoring Instrument (TROPOMI) satellite observations of atmospheric methane for 2018-2023 to quantify emissions on a monthly basis and attribute the causes of the methane increase.

Global methane concentrations increased at a rate of 6-10 ppb a<sup>-1</sup> prior to 2019, surging to 13-18 ppb a<sup>-1</sup> in 2020-2022 before returning to 10 ppb a<sup>-1</sup> in 2023 (NOAA, 2024). The causes of the methane surge are uncertain and have been variably attributed to wetlands or a decrease in OH (Qu et al., 2022; Peng et al., 2022; Qu et al., 2024), with recent work favoring a wetland surge (Drinkwater et al., 2023; Nisbet, 2023; Nisbet et al., 2023; Michel et al., 2024). Earlier increases have been attributed to emissions increases from oil and gas, livestock, and wetlands, with changes in the <sup>13</sup>C-CH<sub>4</sub> isotopic abundance pointing towards a biogenic source (Hausmann et al., 2016; Zhang et al., 2021; Basu et al., 2022; Feng et al., 2023; Zhang et al., 2024). Global daily observations from TROPOMI, launched in 2017 (Lorente et al., 2021), provide a unique dataset to attribute methane trends including seasonal information.

LETKF (Hunt et al., 2007) uses an ensemble of chemical transport model (CTM) simulations of methane concentrations over short successive assimilation time windows to relate emissions to atmospheric concentrations. This ensemble approximates the background error covariance matrix which represents the prior uncertainty in the system. LETKF has been used previously to analyze methane emissions and their trends (Feng et al., 2017; Bisht et al., 2023; Zhu et al., 2022). It has advantages compared to other inverse methods reviewed by Brasseur and Jacob (2017) in being far less computationally expensive than analytical methods, not requiring a model adjoint like 4D-Var methods, and not being restricted dimensionally like Markov chain Monte Carlo methods. The short assimilation time window reduces the effect of errors in model transport (Yu et al., 2021) and in the seasonality of the prior estimate (East et al., 2024).

Here we estimate global methane emissions at 2°×2.5° spatial resolution and monthly temporal resolution from May 2018 through December 2023. We use the CHEEREIO platform (Pendergrass et al., 2023) to apply LETKF to the TROPOMI data. CHEEREIO is a general user-friendly platform for LETKF data assimilation powered by the GEOS-Chem CTM. We use the results to analyze seasonal and 2019-2023 trends in methane emissions from different emission sectors.

### 2 Data assimilation system

We use methane observations from TROPOMI (section 2.1) to optimize global methane emissions at 2°×2.5° resolution (section 2.2) with a LETKF algorithm (section 2.3) implemented through CHEEREIO (section 2.4). We apply a downscaling approach to attribute emissions to different sectors at a finer scale than the  $2^{\circ} \times 2.5^{\circ}$  resolution of the inversion (section 2.5).

#### 2.1 Observations



TROPOMI detects solar backscatter in the 2.3 µm methane absorption band with global daily coverage at 5.5×7 km<sup>2</sup> nadir pixel resolution (7×7 km<sup>2</sup> before August 2019) and 13:30 local solar time. We use the operational retrieval of dry-column methane mixing ratios  $(X_{CH_A})$  from the Netherlands Institute for Space Research (SRON) (Lorente et al., 2023), corrected for bias with a machine-learning algorithm trained on collocated data from the more precise but much sparser GOSAT satellite instrument (Balasus et al., 2023; obtained from https://registry.opendata.aws/blended-tropomi-gosat-methane. Last accessed: 27 Feb 2025).

We filter out retrievals over coastlines (fractional-water pixels) and oceans (glint retrievals), which are subject to residual artifacts (Balasus et al. 2023). We also account for bias that could be introduced by extended periods of missing TROPOMI data, caused by outages of the Visible Infrared Imaging Radiometer Suite (VIIRS) which is used for cloud clearing (Borsdorff et al., 2024). Full TROPOMI data records are available for 2019-2021, but in 2022 no TROPOMI data is available between July 26 and August 23, and in 2023 retrievals begin to fail on July 26 and are fully missing between August 10 and August 30. This is the time of year when northern hemispheric methane concentrations are at their minimum but sharply rising because of wetland missions (East et al., 2024). In the absence of observations, LETKF would persist July emissions through the period of missing data and increase emissions suddenly when observations resume to correct a global bias. We account for this artifact in our

$$x_{\rm yr} = x_{2021} \cdot \frac{x_{\rm yr,valid}}{x_{2021,\rm valid}}$$
 (1)

estimates of interannual variability by scaling to the seasonality of 2021 emissions as follows:  $x_{\rm yr} = x_{\rm 2021} \cdot \frac{x_{\rm yr,valid}}{x_{\rm 2021,valid}} \tag{1}$  Here  $x_{\rm yr}$  are annual posterior mean gridded emissions in yr  $\in$  {2022,2023} after correction,  $x_{\rm yr,valid}$  are annual posterior mean emissions excluding the period of missing data, and  $x_{2021,\text{valid}}$  are 2021 posterior emissions excluding the same period. This assumes similar seasonal variations in the three years. Observed methane concentrations from the NOAA global surface network (NOAA, 2024) show highly reproducible seasonality from year to year (East et al., 2024). The global mean surface concentration in July/August 2021 was 6 ppb below the 2021 annual mean, as compared with 5 ppb in 2022 and 7 ppb in 2023. When analyzing seasonality, we show either 2021 results or the 2019-2021 detrended mean seasonality to avoid bias due to missing observations. We find that the LETKF corrects emissions for the periods of missing data within two 5-day assimilation time windows after TROPOMI observations are available again. There is no need for an extended "burn-in" period, which may be due to our run-in-place methodology which efficiently makes use of available observations (section 2.4). Because we scale emissions gridcell by gridcell, we implicitly account for different emission seasonalities in different regions and latitude bands.

## 2.2 GEOS-Chem, prior inventories, and prescribed methane sinks





GEOS-Chem is a three-dimensional CTM driven by assimilated meteorological data from the Modern-Era Retrospective analysis for Research and Applications, Version 2 (MERRA-2) of the NASA Global Modeling and Assimilation Office (GMAO). We use the GEOS-Chem methane simulation (Maasakkers et al., 2019) at 2.0°×2.5° resolution. We initialize all ensemble members in 2018 with a 33-year GEOS-Chem simulation in which the methane field is controlled by time-varying gridded NOAA surface methane observations that are used as the simulation's lower boundary condition, thus properly initializing the stratosphere (Mooring et al., 2024).

Prior methane emissions are listed in **Table 1**. Prior estimates of emissions and loss include no trends over the study period (persisting 2019 values), so that any trends in the posterior solution are due to observations. Anthropogenic emissions are assumed to be aseasonal, except for manure management and rice for which we apply seasonal scaling factors (Maasakkers et al., 2016; Zhang et al., 2016a). For wetland emissions, we conduct parallel inversions with prior estimates based on two alternative inventories: the mean of the nine-member high-performance subset of the WetCHARTs v1.3.1 inventory ensemble (Bloom et al., 2017; Ma et al., 2021), and the Lund–Potsdam–Jena Wald Schnee und Landschaft (LPJ-wsl) dynamic global vegetation model driven with assimilated meteorological data from MERRA-2 (Zhang et al., 2016b). The latter inventory, which we denote LPJ-MERRA2 in what follows, was found by East et al. (2024) to uniquely match the observed global methane seasonality as compared to other wetland emission inventories (East et al., 2024). As discussed later, many emission sources are co-located making source attribution difficult, especially in eastern Africa where livestock and wetlands overlap substantially.

Table 1. Global methane sources (Tg a<sup>-1</sup>) for 2023

|               | Prior estimate <sup>a</sup> | Posterior best estimate <sup>b</sup> |
|---------------|-----------------------------|--------------------------------------|
| Total         | 529-574                     | 570                                  |
| Anthropogenic | 348                         | 392                                  |
| Livestock     | 121*                        | 151                                  |
| Oil+Gas       | $50^{\dagger}$              | 60                                   |
| Coal          | $34^{\dagger}$              | 26                                   |
| Rice          | 39*                         | 36                                   |
| Waste         | 81*                         | 92                                   |
| Other         | 24*                         | 26                                   |
| Natural       | 181-226                     | 178                                  |
| Wetlands      | 148-193                     | 141                                  |
| Termites      | 12                          | 18                                   |
| Fires         | 19                          | 17                                   |
| Seeps         | 2                           | 2                                    |

<sup>&</sup>lt;sup>a</sup>Prior emissions include no trends over 2018-2023. Ranges are defined by the two alternative prior estimates for wetlands, both at 0.5°×0.5° monthly resolution for 2019: lower value is WetCHARTs v1.3.1 (Ma et al., 2021) higher value is LPJ-wsl driven by MERRA-2 meteorology (Zhang et al., 2016b). Prior non-fossil anthropogenic emissions are from the 2018 EDGARv6 inventory (Crippa et al., 2021), denoted \*, and fossil anthropogenic emissions are from the 2010-2019 Global Fuel Exploitation Inventory (GFEI) version 2.0 (Scarpelli et al., 2022), denoted †. All anthropogenic emissions are at 0.1°×0.1° resolution and are overwritten by national gridded emissions for the contiguous US

(Maasakkers et al., 2016), Mexico (Scarpelli et al., 2020), and Canada (Scarpelli et al., 2021). Termite emissions  $(4^{\circ}\times5^{\circ})$  are from Fung et al. (1991), fire emissions  $(0.25^{\circ}\times0.25^{\circ})$  are from the 2019 Global Fire Emissions Database (GFED4) (van der Werf et al., 2017), and geological seeps  $(1^{\circ}\times1^{\circ})$  are from Etiope et al. (2019) with global scaling to the annual total from Hmiel et al. (2020).

<sup>b</sup>Posterior emissions for 2023 from the LETKF with sources attributed via downscaling. Best estimate represents the mean of LPJ-MERRA2 and WetCHARTs posterior estimates both with and without methane concentrations in the state vector.

Loss of methane from oxidation by tropospheric OH is computed with global 3-D monthly mean OH fields from GEOS-Chem (Wecht et al., 2014), scaled so that methane's steady-state lifetime due to loss to tropospheric OH matches the best estimate of 11.2 years derived from methyl chloroform observations (Prather et al., 2012; East et al., 2024). We assume no interannual variability in tropospheric OH concentrations. Additional minor methane sinks in GEOS-Chem include oxidation by tropospheric Cl (Wang et al., 2019), oxidation in the stratosphere (Mooring et al., 2024), and uptake by soils (Murguia-Flores et al., 2018), resulting in an overall methane lifetime of 9.4 years.

We do not optimize tropospheric OH concentrations (as the main methane sink) because they do not imprint local gradients of methane concentrations as needed for application of LETKF. Global analytic inversions optimize OH concentrations independently of emissions by exploiting knowledge of the global OH distribution (Zhang et al., 2018; Maasakers et al., 2019; Penn et al., 2025). Interannual variability of OH concentrations may in fact contribute to interannual variability of methane concentrations (Bouarar et al., 2021; Peng et al., 2022; Morgenstern et al., 2025), but emission changes are more important (Feng et al., 2023; Qu et al., 2024; He et al., 2025).

## 2.3 The LETKF algorithm




The LETKF algorithm optimizes a state vector of emissions, or of concatenated emissions and concentrations, to minimize the Bayesian scalar cost function J(x) assuming Gaussian error probability density functions (pdfs; Hunt et al., 2007; Brasseur and Jacob, 2017):

$$J(x) = (x - x^b)^T (P^b)^{-1} (x - x^b) + \gamma (y - H(x))^T R^{-1} (y - H(x))$$
 (2)

Here x is the state vector to be optimized,  $x^b$  is the prior estimate,  $P^b$  is the background (also called prior or forecast) error covariance matrix of the model prediction, y is the TROPOMI observations,  $H(\cdot)$  is an observation operator that transforms the state vector x from the state space to the observation space, R is the observational error covariance matrix, and y is a regularization constant to account for unresolved error correlation in the observations and is taken to be 0.1 following Qu et al., (2024). x includes gridded  $2^{\circ} \times 2.5^{\circ}$  methane emission scaling factors over land excluding Antarctica (2737 state vector elements) to be applied to the prior estimates. In additional inversion ensemble runs, we simultaneously optimize methane concentrations along with emissions scaling factors (concatenating both in the state vector x) to avoid systematically attributing discrepancies between observations and background concentrations to errors in prior emissions.

In the LETKF, m ensemble members with different emissions are initialized at time  $t_o$  and the forward model (GEOS-Chem) is run in parallel for a user-specified time (termed the assimilation window) for each of these ensemble members. After the runs complete, we construct the state vectors  $\mathbf{x}_i^b$  for each ensemble member (indexed by i). We localize the calculation within a 1500 km radius of the grid cell being optimized, considering only observations within that radius; this converts a single intractable large

matrix problem into many embarrassingly parallel calculations for individual grid cells involving much smaller matrices. We weight observations by their distance from the target grid cell with the Gaspari-Cohn function, a piecewise polynomial resembling a bell curve with a value of 1 at the grid cell and 0 at 1500 km away (Gaspari and Cohn, 1999).

To optimize the methane emissions, or concatenated emissions and concentrations of a given grid cell, we start from the background state vector  $x_i^b$ , and form the background perturbation matrix  $X^b$  from the m vector columns  $X_i^b$ :

$$X_{i}^{b} = x_{i}^{b} - \overline{x^{b}}; \ \overline{x^{b}} = \frac{1}{m} \sum_{i=1}^{m} x_{i}^{b}$$
 (3)

Here  $X_i^b$  represents the *i*th column of the  $n \times m$  matrix  $X^b$  where n is the length of the state vector; each column of  $X^b$  consists of the state vector from an ensemble member minus the mean state vector. The model predictions made during the assimilation window must be compared to observations. Hence we construct background vectors of simulated observations  $y_i^b$  and a corresponding simulated observation perturbation matrix  $Y^b$  formed from the m vector columns  $Y_i^b$ :

$$Y_{i}^{b} = y_{i}^{b} - \overline{y^{b}}; \quad y_{i}^{b} = H(x_{i}^{b}); \quad \overline{y^{b}} = \frac{1}{m} \sum_{i=1}^{m} y_{i}^{b}$$
 (4)

All simulated observations are timed to line up as close as possible with actual observations (in this case, within one hour).

The mean analysis (posterior) state vector in the original space is then given by (Hunt et al., 2007):

$$\overline{x^a} = \overline{x^b} + \gamma X^b \widetilde{P^a} (Y^b)^T R^{-1} (y - \overline{y^b})$$
 (5)

where y is the vector of observations.  $\widetilde{P}^{a}$  is an  $m \times m$  matrix computed as follows:

$$\widetilde{\mathbf{P}^a} = \left( \left( (m-1) \cdot \mathbf{I} \right) + \gamma \left( \mathbf{Y}^b \right)^T \mathbf{R}^{-1} \mathbf{Y}^b \right)^{-1} \tag{6}$$

where I is the  $m \times m$  identity matrix. The analysis perturbation matrix is then given by

$$X^{a} = X^{b} \left( (m-1)\widetilde{P}^{a} \right)^{\frac{1}{2}} \tag{7}$$

From here, the new ensemble state vectors can be constructed by adding  $\overline{x^a}$  back to each column of  $X^a$ . With the ensemble updated and errors characterized, the ensemble can be evolved using GEOS-Chem for the next assimilation window.

# 5 2.4 The CHEEREIO platform and LETKF settings


CHEEREIO is a lightweight wrapper of GEOS-Chem written in Python which allows users to conduct a range of LETKF applications by editing a single configuration file (Pendergrass et al., 2023). It takes advantage of GEOS-Chem's HEMCO emission module to update emissions without modifying the source code (Lin et al., 2021). Here we describe several new settings in the CHEEREIO v1.3.1 implementation of LETKF (<a href="https://doi.org/10.5281/zenodo.11534085">https://doi.org/10.5281/zenodo.11534085</a>), which we use in this work.

**Figure 1** shows the LETKF workflow as implemented in CHEEREIO v1.3.1. We apply the run-in-place (RIP) method to the LETKF assimilation window (Kalnay and Yang, 2010; Liu et al., 2019).

With RIP, we calculate the LETKF assimilation update using a long period of observations (15 days, called the observation window), but then advance the assimilation window forward for a shorter period (5 days). RIP thus maintains linear growth in posterior perturbations and allows the system more time to assimilate information. Importantly, after advancing the assimilation window forward, we do not reinitialize the ensemble for new runs. Instead, the assimilated state of the previous observation window becomes the initial background state of the next assimilation window.

Figure 1: Flowchart of CHEEREIO v1.3.1 LETKF inversion procedure for assimilating TROPOMI methane data. We initialize a GEOS-Chem CTM simulations with randomized multiplicative perturbations to the prior estimates, applied to each of the 32 ensemble members. For assimilation period *k*, CHEEREIO runs GEOS-Chem for the observation window (15 days) for each ensemble member, then conducts the LETKF inversion by comparing the ensemble of GEOS-Chem values to the TROPOMI observations, over the observation window. Posterior emission scaling factors and concentrations are then inflated to reflect the prior spread using the RTPS procedure. The posterior emission estimates and inflated concentrations then become the prior estimate for the *k*+1 assimilation period advancing by 5 days.

Because ensemble-based methods undersample the prior probability space, they suffer from shrinking dispersion between ensemble members which can lead to artificially small prior error estimation; an error inflation method is necessary to prevent ensemble collapse (Hunt et al., 2007). Following Bisht et al. (2023), we use the Relaxation to Prior Spread (RTPS) inflation method (Whitaker and Hamill, 2012). RTPS inflates the posterior ensemble standard deviation  $\sigma^a$  (defined as the standard deviation of each state vector element) of such that it partially reflects the background ensemble standard deviation  $\sigma^b$ :







$$X_{\text{infl}}^{a} = \left(\frac{\alpha_{\text{RTPS}}\sigma^{b} + (1 - \alpha_{\text{RTPS}})\sigma^{a}}{\sigma^{a}}\right)X^{a}$$
 (8)

Here  $\alpha_{RTPS}$  is a parameter between 0 and 1 which represents the weighted contribution of the background standard deviation  $\sigma^b$  in inflating the analysis ensemble to obtain the final analysis perturbation matrix  $X_{infl}^a$ . After sensitivity tests to mitigate underdispersed ensemble spread (shown by decreasing fidelity to observations over time), we take  $\alpha_{RTPS}$  to be 0.7, which is consistent with optimized values in Bisht et al. (2023). In the runs where only methane emissions are optimized, we additionally apply RTPS to 3D methane concentrations in the ensemble members even though we do not formally include concentrations in the state vector.

We perform our emissions estimates with an assumption of lognormal errors on the prior emission estimates, as is commonly done for analytical inversions (Maasakkers et al., 2019; Hancock et al., 2025) but to our knowledge has not previously been applied in the LETKF formalism. A lognormal error pdf better captures the upper tail of the methane emissions distribution than normal errors (Duren et al., 2019; Cusworth et al., 2022) and also prevents unphysical negative posterior emission estimates (Miller et al., 2014) considering that we do not optimize the soil sink. However, a lognormal distribution across ensemble members violates the assumptions of the LETKF equations (Hunt et al., 2007). We solve this problem by sampling methane emissions scaling factors for each ensemble member according to a lognormal distribution centered on 1 (prior emission inventory) and run GEOS-Chem for each ensemble member with these scaling factors applied. When it is time for the LETKF calculation, we apply a logarithmic transform to the methane scaling factor distributions and thus obtain a normal distribution (centered on 0) for the construction of the background perturbation matrix  $X^b$ . We perform the LETKF and once it is complete we apply an exponential to transform back to the original lognormal distribution, which is then used to evolve GEOS-Chem once more. These transformations are indicated as "log" and "exp" in Figure 1. The posterior solution is then the median of the LETKF ensemble.

Before ingesting the TROPOMI observations into the LETKF, we aggregate the original observations into "super-observations" by averaging them onto the  $2.0^{\circ}\times2.5^{\circ}$  GEOS-Chem grid (Eskes et al., 2003; Miyazaki et al., 2012; Pendergrass et al., 2023; Chen et al., 2023). To model the reduction in observational error variance due to averaging and obtain the super-observation error standard deviation  $\sigma_{\text{super}}$ , we follow a two-component error variance equation which separates contributions due to forward model transport error variance ( $\sigma_{\text{transport}}^2$ ) and error variance for a single retrieval ( $\sigma_i^2$ ):

$$\sigma_{\text{super}} = \sqrt{\left[\left(\frac{1}{p}\sum_{i=1}^{p}\sigma_{i}\right)\cdot\left(\frac{1-c}{p}+c\right)\right]^{2} + \sigma_{\text{transport}}^{2}}$$
(9)

Here p is the number of observations aggregated into a super-observation and c is the error correlation between the individual retrievals within a super-observation. The transport error is fully correlated. We take  $\sigma_i = 17$  ppb,  $\sigma_{transport} = 6.1$  ppb, and c = 0.28 based on an empirical residual error method fit for TROPOMI methane (Chen et al., 2023; Pendergrass et al., 2023).

## 255 **2.5 Sub-grid source attribution**





Our inversion optimizes emissions on a  $2^{\circ}\times2.5^{\circ}$  grid but the bottom-up inventories and TROPOMI data have much finer resolution ( $0.1^{\circ}\times0.1^{\circ}$  for anthropogenic emissions,  $0.5^{\circ}\times0.5^{\circ}$  for wetland emissions,  $7\times7$  km² or  $5.5\times7$  km² for TROPOMI observations at nadir). Here we exploit this high-resolution data with the source attribution approach of Yu et al. (2023), in which we conserve the overall posterior emissions in a  $2^{\circ}\times2.5^{\circ}$  grid cell but adjust relative source contributions within it based on subgrid observational patterns. If TROPOMI observations are persistently elevated in a portion of the  $2^{\circ}\times2.5^{\circ}$  grid cell associated with a particular sector, the Yu et al. (2023) methodology will attribute a larger fraction of the correction to that sector. We neglect subgrid prior error terms in Yu et al. (2023) to obtain a subgrid attribution based solely on the distribution of TROPOMI observations and prior sources. For wetlands we update the prior sources for individual years using LPJ-MERRA2.

Most grid cells are not affected significantly by this sub-grid source attribution approach, but we find substantial adjustments in a few regions including Sudd wetlands in South Sudan (where some livestock emissions are re-attributed to wetlands) and in Bangladesh (where some rice emissions are re-attributed to wetlands). Our global posterior wetlands emission increases by 10%, offset by decreases in the rice, livestock, and waste sectors. In some regions, especially eastern Africa, estimates of livestock emissions are highly uncertain, so we will make use of additional data sources in our interpretation of results below.

#### 3 Results and discussion

Figure 2 shows TROPOMI methane dry column mixing ratios (XCH<sub>4</sub>) for the study period, along with the corresponding GEOS-Chem model biases using prior and posterior emissions with either WetCHARTs or LPJ-MERRA2 as prior emissions for wetlands. The model with prior emissions has a low bias due to a methane budget imbalance. The posterior emissions eliminate this bias. Figure 3a shows the growth in global annual mean methane concentrations over the 2018-2023 study period. Trends in NOAA surface methane concentrations (NOAA, 2024) are consistent with TROPOMI trends as well as our posterior estimate. Figure 3b shows the posterior emissions from our four inversion ensemble members (driven with different wetlands and either optimizing concentrations and emissions or emissions alone), all predicting similar annual emissions (577 Tg a<sup>-1</sup> and 567 Tg a<sup>-1</sup> for WetCHARTs and LPJ-MERRA2 respectively in 2023). Seasonal CH<sub>4</sub> variability and trends in both hemispheres are also well-captured by the posterior (Figure 3cd).

Posterior emissions for 2023 are summarized in Table 1. Our best posterior estimate of 560 Tg a<sup>-1</sup> for 2019 is within the 556-570 Tg a<sup>-1</sup> range calculated for 2019 by Qu et al. (2021) and the 553–586 Tg a<sup>-1</sup> range from top-down inversions for 2010-2019 reviewed in Saunois et al. (2025). In 2020 and 2021

we find that global methane emissions surged to 587 Tg a<sup>-1</sup> and 592 Tg a<sup>-1</sup> before declining to 572 Tg a<sup>-1</sup> in 2022 and 570 Tg a<sup>-1</sup> in 2023, consistent with the 570-590 Tg a<sup>-1</sup> range for 2020-2022 reported in Ou et al. (2024) and the 2020 estimate of 581-627 Tg a<sup>-1</sup> from Saunois et al. (2025). Our best estimate of fossil 290 fuel emissions (oil, gas, and coal) for 2019 is 88 Tg a<sup>-1</sup>, intermediate between the 80 Tg a<sup>-1</sup> found in the 2019 analytical inversion of Ou et al. (2021) and the 98 Tg a<sup>-1</sup> estimate for 2018-19 from 4D-Var inversions done by Yu et al. (2023), but below the 100-124 Tg a<sup>-1</sup> range for 2010-2019 in Saunois et al. (2025). Our 265 Tg a<sup>-1</sup> estimate for agricultural and waste emissions for 2019 is correspondingly above the 213-242 Tg a<sup>-1</sup> range for 2010-2019 in Saunois et al. (2025), while our wetland posterior estimate of 150 Tg a<sup>-1</sup> falls within but at the low end of the 145-214 Tg a<sup>-1</sup> range from Saunois et al. (2025) and is lower than Qu et al. (2024). We do not account for interannual variability in tropospheric OH, the main methane sink; a changing sink would impact inferred emissions by mass balance. In particular, if OH concentrations declined in 2020 due to COVID-19 lockdowns and increased biomass burning emissions, mass balance would imply a smaller methane emissions surge than found here (W. Chen et al., 2025). However, we predict similar posterior emissions as previous studies which do optimize OH (Ou et al., 2024: He et al., 2025).

# TROPOMI observations and model bias using prior and posterior emissions, 2018-2023

Figure 2: TROPOMI observations of methane dry column mixing ratios (XCH<sub>4</sub>) and comparison to GEOS-Chem simulations using either prior or posterior emissions. Values are averages for June 2018 through December 2023. ΔCH<sub>4</sub> denotes the difference between the simulation (with observation operators applied) and the observations. Global mean bias and spatial standard deviation are given inset. Results are shown for wetlands prior estimates from either LPJ-MERRA2 or WetCHARTs.

Figure 3: Global methane trends, 2018-2023. Panel (a) shows global annual mean observations from NOAA background surface sites (<a href="https://gml.noaa.gov/ccgg/trends\_ch4/">https://gml.noaa.gov/ccgg/trends\_ch4/</a>), TROPOMI, and GEOS-Chem model simulations using prior emission estimates (including either WetCHARTs or LPJ-MERRA2 wetlands) and posterior emission estimates. The posterior represents the mean of the inversion ensemble. Panel (b) shows annual posterior methane emissions for the inversion ensemble, including either WetCHARTs or LPJ-MERRA2 wetlands and either with or without optimization of concentrations. Panels (c) and (d) show mean TROPOMI and GEOS-Chem results smoothed over monthly temporal resolution for the northern (c) and southern (d) hemispheres.

To understand the drivers of our posterior emissions trends, we disaggregate our results by region and sector (Figure 4). We find a negative trend in South American emissions which we attribute to a decline in wetland emissions; this is consistent in sign with other top-down work using GOSAT and surface observations finding decreases in 2020 and 2021 relative to 2019 in the Orinoco, Pantanal, and Amazon Basin wetlands (Lin et al., 2024). We attribute the 2020 methane surge to a 14 Tg a<sup>-1</sup> increase in emissions from sub-Saharan Africa, as in previous studies (Qu et al., 2022; Feng et al., 2022), and we find that the elevated emissions persist into later years. Consistent with Qu et al. (2024), who find that wetland emissions are relatively constant over 2019-2022 and that anthropogenic emissions drive much of the 2020-2021 surge, we find that a surge in wetland emissions contributed to the 2020-2021 emissions peak but anthropogenic sectors including livestock and waste are more important (Figure 4b).




However, anthropogenic attribution of the African emission surge may be unreliable given uncertainty in tropical wetland prior inventories. Figure 5 compares our posterior emissions for the northern tropics and boreal latitudes with water storage from inundation as measured by the Gravity Recovery and Climate Experiment Follow-On (GRACE-FO) twin satellites, where the distance between the satellites is used to measure liquid water equivalent (LWE) thickness anomalies (cm) relative to a

time mean at monthly 0.5°×0.5° resolution (Watkins et al., 2015; Wiese et al., 2016; Wiese et al., 2023). The northern tropics (0°N-30°N) explain much of the 2020-2021 surge and this corresponds closely with increases in water storage; consistently, the declining emission trend in boreal regions (50°N-90°N) corresponds with drying. Much of the northern tropics surge is associated with wetlands in South Sudan and southern Sudan, which account for 9% of prior emissions (mean of WetCHARTs and LPJ-MERRA2) in the 0°N-30°N band but for our posterior 2021-23 estimate they surge to almost a third; indeed, we find a 7.5 Tg a<sup>-1</sup> increase from 2019 to 2021 in the region, accounting for a quarter of the global emissions increase in the same period. Our posterior solution predicts sharply increasing emissions after 2019 in the Sudd, Machar, and Lotilla wetlands in South Sudan, which experienced extensive flooding in 2020 and in following years and have been identified in previous work as globally significant drivers of the methane emissions trend (Pandey et al., 2021; Feng et al., 2022; Hardy et al., 2023). Flooding of these areas is associated with anomalous rainfall in the "short rains" season, driven by a strongly positive Indian Ocean Dipole (IOD) event, with warmer ocean surface temperatures in the western Indian ocean driving convection (Wainwright et al., 2021; Lunt et al., 2021, Palmer et al., 2023). There is some consensus in climate models that both precipitation during the short rains and the frequency of extremely positive IOD events will increase with climate change (Palmer et al., 2023), supporting the interpretation of this tropical methane emission surge as a positive climate feedback. We attribute almost half of the 7.5 Tg a<sup>-1</sup> increase in Sudan and South Sudan to anthropogenic sources (principally livestock) but this may reflect an underestimate of wetland area in the prior inventories. Although it is difficult to separate livestock and wetland emissions in this region due to co-location and isotopic similarities, additional data sources capturing changes in inundation can offer evidence for wetland emissions over livestock.





Recent work indeed suggests that wetland extent in Africa may be underestimated due to sparse observational data (Dong et al., 2024), and methane emissions from vegetated tropical wetlands may more generally be underestimated by mechanistic models (France et al., 2022; Shaw et al., 2022). Wetlands in South Sudan especially are prone to underestimates from wetland models because emissions are driven by inflows from the White Nile and Sobat rivers rather than local precipitation (Pandey et al., 2021). The post-2020 period corresponds with record high water levels in Lake Victoria which feeds the White Nile, with the short rains at the end of 2019 driving a 1.5 m increase in lake water levels; water levels rose at a downstream river station through 2022 even after Lake Victoria water levels began to fall (Dong et al., 2024). High water levels in the Blue Nile also slow White Nile discharge, further contributing to sustained flooding (Smith and El-Kammash, 2025). Neither the WetCHARTs nor the LPJ-MERRA2 inventories capture the surge in these wetlands. As a result, our inversion and the previous inversion of Qu et al. (2024) attribute the 2019-2021 methane surge to a 40% increase in livestock emissions in sub-Saharan Africa, While livestock populations have grown (Nisbet et al., 2025), such an increase is inconsistent with Food and Agricultural Organization (FAO) cattle population data, which shows only an increase of 8% in 2023 relative to 2019 in the region (https://www.fao.org/faostat; last accessed: 2025-02-07). As Figure 6 shows, total emissions for the region including the increase after 2019 and seasonal emission peak are closely associated with GRACE-FO water storage data, while emissions attributed to wetlands in WetCHARTs or LPJ-MERRA2 do not reflect GRACE-FO trends. Our methodology is able to detect methane surges in these wetlands in part because they are river-fed and less obscured by cloud cover than other wetlands, such as the Congo and the Amazon, which are difficult for solar backscatter retrievals to observe especially in the rainy season (Figure 2b).

We see from Figure 6 that inundation as measured by GRACE-FO is strongly correlated with the seasonality of methane emissions in sub-Saharan Africa. Figure 7 shows the global seasonal cycle of posterior methane emissions for 2021, avoiding missing TROPOMI observations in 2022-23. The global seasonality of methane emissions is mainly driven by the northern hemisphere. The seasonality of methane in the southern hemisphere (Figure 3) is largely driven by the OH sink (East et al., 2024). Unlike the prior estimates including WetCHARTs or LPJ-MERRA2 wetlands, which show a July-August peak in the northern hemisphere (Figure 7a) in line with other wetland models (Zhang et al., 2024), we find a sharp September peak driven by tropical emissions which strongly influences global seasonality (Figure 7b). Figure 7c shows that the peak of northern tropical emissions corresponds with the peak of mean GRACE-FO water storage data, and occurs later in the year than implied by prior inventories. Livestock shows a seasonality in phase with wetlands, which as pointed out above could be due to misattribution in the tropics, though food availability for cattle in eastern Africa may be in phase with wetland extent. Rice emissions in the northern hemisphere peak in July-September corresponding to the dominant growing season (Z. Chen et al., 2025) and may increase in importance as rice production increases in Africa (Chen et al., 2024).

Figure 4: Annual emission trends for 2019-2023 disaggregated by region and sector. Panels (a) and (b) show posterior emission changes relative to 2019, disaggregated by region and sector respectively. Inset percentages show changes relative to 2019 values for selected regions/sectors. Error bars show range of inversion ensemble for the global emission trend. Panel (c) shows 2019-2023 trends in posterior emissions by region obtained from linear regression.

# Posterior emission and inundation annual trends

Figure 5: Inundation and posterior emission trends. Top panel compares total posterior emission north of 50°N with mean GRACE-FO liquid water equivalent (LWE) anomalies weighted by gridded total posterior emissions. Bottom panel is the same but for the northern tropics (0-30°N).

Figure 6: Inundation and posterior emissions trends in sub-Saharan Africa (region defined in Figure 4). The panel compares total and wetlands posterior emissions with mean GRACE-FO liquid water equivalent (LWE) anomalies. LWE is an average for the region, weighted by gridded total posterior emissions.

Figure 7. Posterior emission seasonality in 2021. Panel (a) shows northern hemisphere posterior emissions disaggregated by source sector, where the seasonal cycle is obtained by subtracting the 2021 mean. The prior seasonal cycle of total emissions is also shown in grey lines for both LPJ-MERRA2 and WetCHARTs wetlands. Panel (b) is as in panel (a) but global and disaggregated by latitude. Error bars show range of inversion ensemble. Panel (c) shows prior simulations driven by LPJ-MERRA2 and WetCHARTs wetlands with the posterior best estimate in the northern tropics (0°N-30°N), compared to mean GRACE-FO liquid water equivalent (LWE) anomalies in the region, weighted by gridded total posterior emissions.


Fossil fuel emissions are generally considered to be aseasonal, with the possible exception of Russian pipelines (Reshetnikov et al., 2000), but we observe seasonality in some production basins especially in the US. Figure 8 shows the difference in best-estimate posterior fossil fuel emissions in cold months minus warm months, with many areas showing elevated cold season emissions including several major US basins, Hassi R'Mel field (Algeria), Sirte basin (Libya), and West Karun basin (Iran). This phenomenon has been observed before in the Permian (Vanselow et al., 2024; Hu et al., 2025; Varon et al., 2025), but it is not seen worldwide and may suggest processes specific to the industry in the US and a few other regions. Possible causes include more frequent equipment failures in winter or emissions from poorly weatherized separator vessels, where more gas remains dissolved in liquid at cold temperatures and is vented later from liquids storage tanks (Varon et al., 2025). We do not find oil and gas seasonality in Russia or other boreal regions, but this may be due to poor observational capacity in the winter.

# Cold minus warm season fossil emissions

Figure 8: Seasonality of oil and gas emissions for 2019-2021. Panel (a) shows northern hemisphere mean fossil fuel emissions in cold months (December through April) minus warm months (June through September) in grid cells where fossil fuels account for at least 50% of emissions. Inset (b) is as in (a) but for the contiguous US (CONUS), with major sedimentary basins overlaid; inset (c) is as in (a) but for Algeria and Libya, with oil and gas fields overlaid (Sabbatino et al., 2017).

#### **4 Conclusions**



We used the localized ensemble transform Kalman filter (LETKF) algorithm, deployed through the open-source CHEEREIO platform, to infer global methane emission trends and seasonalities by assimilation of TROPOMI satellite observations of atmospheric methane from May 2018 through December 2023 over 5-day time windows. Our goal was to understand the regions and source sectors driving the rapid increase of methane over that period and its seasonality. We used the blended TROPOMI product of Balasus et al. (2023) that corrects TROPOMI retrieval biases using machine learning applied to collocated observations from the GOSAT satellite instrument.

Our posterior emissions from the assimilation of TROPOMI data reproduce the observed 2019-2023 trends in methane concentrations at surface sites and from TROPOMI, with minimal regional bias. We estimate that emissions surged from 560 Tg a<sup>-1</sup> in 2019 to 587-592 Tg a<sup>-1</sup> in 2020-2021 before declining to 572-570 Tg a<sup>-1</sup> in 2022-2023, not accounting for possible changes in OH concentrations. Sub-Saharan Africa contributed 14 Tg a<sup>-1</sup> of the 27 Tg a<sup>-1</sup> global increase in 2020 and this contribution was sustained through 2023. Past attribution of this surge to anthropogenic sources may be due to errors

in the spatial distribution of wetlands, as we find that the emission increases correspond closely with inundation as measured by the GRACE-FO satellite instrument. Wetlands in East Africa, particularly the Sudd, are instrumental in driving the methane trend but are poorly represented in current wetland emission models.

Methane emissions show a large seasonality and the high temporal resolution of our LETKF implementation allows us to probe its origin. We find that this seasonality is dominated by northern hemisphere wetland emissions and peaks in September, as opposed to July in wetland models. The September peak in the tropics closely follows inundation patterns. This finding is in line with previous work showing a mismatch between field observations and tropical wetland emissions predicted by models, and points towards the need for improved modelling in this critical region. Oil and gas emissions show little seasonality globally but we find that production fields in the US have a distinct seasonal cycle of elevated emissions during the cold season.

Limitations of this study include remaining uncertainties regarding source attribution and possible variability in OH concentrations. Eastern Africa is a key region driving the methane emission trend, but the spatial and seasonal variability of livestock and wetland emissions in this region are highly uncertain and local observations are lacking. While GRACE-FO indicates wetland inundation as an emissions driver, future work could improve source attribution by improving prior emissions inventories, especially wetland models, Our approach also does not consider the effects of changing methane loss to OH over 2019-2023. If tropospheric OH declined in 2020, it could imply a smaller methane emissions surge than found here. Incorporation of oxidation products like CO and formaldehyde may help constrain OH (Yin et al., 2021).

Our CHEEREIO software toolkit is openly available (https://doi.org/10.5281/zenodo.11534085) as a general user-friendly implementation of LETKF for assimilating observations of atmospheric composition through the GEOS-Chem chemical transport model. In this work we introduced a novel approach to specify log-normal emissions errors within the LETKF framework, and this is released as part of CHEEREIO version 1.3.1.

# Data and code availability







The CHEEREIO source code is available at https://github.com/drewpendergrass/CHEEREIO; the version of CHEEREIO used in this paper (1.3.1) is archived at https://doi.org/10.5281/zenodo.11534085 (Pendergrass et al., 2023). GEOS-Chem version 14.1.1 source code is archived https://doi.org/10.5281/zenodo.7696632. The blended TROPOMI-GOSAT product is available at https://registry.opendata.aws/blended-tropomi-gosat-methane (Balasus et al., 2023) and NOAA surface data is available at (https://gml.noaa.gov/ccgg/trends ch4/). Wetland emissions from WetCHARTs v1.3.1 are available at https://doi.org/10.3334/ORNLDAAC/1915 (Ma et al., 2021) and from LPJ-wsl at https://gmao.gsfc.nasa.gov/gmaoftp/lott/CH4/wetlands/. Oil, gas, and coal emissions from the GFEIv2 inventory are available at https://doi.org/10.7910/DVN/HH4EUM and other anthropogenic emissions are available from EDGARv6 at https://doi.org/10.2760/074804. Regional anthropogenic emissions are available for the contiguous US (https://www.epa.gov/ghgemissions/gridded-2012-methane-emissions), (https://doi.org/10.7910/DVN/CC3KLO). Canada and Mexico (https://doi.org/10.7910/DVN/5FUTWM). GRACE-FO data are from https://doi.org/10.5067/TEMSC-3JC634. Scaled OH fields, the stratospheric-adjusted GEOS-Chem restart file, stratospheric loss rates, CHEEREIO configuration files, and base HEMCO configuration file required to reproduce this work are permanently archived on Zenodo at https://doi.org/10.5281/zenodo.15120760. Monthly gridded posterior emissions for the posterior best estimate is also provided at https://doi.org/10.5281/zenodo.15120760.
 Additional data related to this study can be obtained on request.

## Acknowledgements

This work was supported by the NASA Carbon Monitoring System. DCP was funded in part by an NSF Graduate Research Fellowship Program (GRFP) grant. This work was funded in part by an appointment to the NASA Postdoctoral Program at the Jet Propulsion Laboratory, California Institute of Technology, administered by Oak Ridge Associated Universities under contract with NASA.

## **Competing interests**




The corresponding author has declared that none of the authors has any competing interests.

## **Author contributions**

DCP and DJJ designed the study. DCP built the CHEEREIO v1.3.1 platform and performed the inversion with contributions from NB, LE, DJV, JDE, MH, TAM, EP, and HN. NB provided the TROPOMI-GOSAT product and offered guidance. DJJ, NB, LE, DJV, JDE, MH, TAM, EP, HN, and JRW discussed results and interpretation. DCP and DJJ wrote the paper with input from all authors.

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
