# Peer review of "Trends and seasonality of 2019–2023 global methane emissions inferred from a localized ensemble transform Kalman filter (CHEEREIO v1.3.1) applied to TROPOMI satellite observations"

_EGUsphere, 2025_

## Author Response (AR1)

**Reviewer 1**

**Pendergrass et al. Trends and Seasonality of 2019-2023 global methane emissions. General Comments**

This is an important paper that should be published, but after revision. The topic is of major interest, both scientifically and to wider society, since the dramatic rise in the atmospheric methane burden during the 2019-2023 period was very unexpected and poses a major challenge to the hopes of the UN Paris Agreement and to the Global Methane Pledge. The paper is scientifically innovative, using satellite data and impressive new modelling methodology to assess emissions, and the findings, that tropical wetlands played a major role, are consistent with other studies. However I have some concerns about details of the work, and recommend minor revision before final publication.

**Response**: We thank the reviewer for their support of our work, and for their helpful comments which have improved the manuscript. All line numbers refer to the clean revised manuscript without tracked changes.

**Wider comments**

The manuscript makes no attempt to discern a wider cause of the growth. For example, Lunt et al 2021, Palmer et al 2023 and Nisbet et al 2023a all discussed the Indian Ocean Dipole. These are large events, unusual and important. It would be nice to address the deeper meterological causes.

**Response**: We think the reviewer for pointing out this issue, and have modified the manuscript throughout to pay more attention to the meteorological driver of the eastern Africa methane emission surge. In particular, lines 338-344, 355-358, and 366-369 among others have been added to link the short rain surge to a positive IOD event, discuss the connection between climate change and changes in rainfall in the region, quantify the extent of flooding in 2019-2020, and discuss how meteorological patterns affect retrievals.

**Overall conclusion**

This is a very interesting and potentially important paper that should be published, but it needs some revision.

**Specific Points**

Line 32- "previous work has attributed the surge to human-caused emissions rather than wetland." That's not fully correct – see for example Drinkwater et al., 2023, who "suggest that wetlands have played a significant role in recent growth of atmospheric CH4". Similar conclusions were reached by Nisbet et al. 2023a,b, and Michel et al. 2024, neither of whom are cited. Similarly, discussing a slightly earlier time frame, there are many papers such as Zhang et al. 2024, and Feng et al. 2023 (already cited elsewhere in the manuscript).

**Response**: We appreciate the reviewer bringing up this point. In lines 31-32 we narrow claim to refer to certain kinds of source attribution. We reviewed the suggested papers and agree that they offer are important to cite. In lines 51-52 we now explicitly highlight

the emerging agreement on wetlands, citing Drinkwater2023, Nisbet2023ab, Michel2024. In our existing sentence on the earlier timeframe (lines 54-55) we now also cite Zhang2024 and Feng2023.

Lines 50-55 – should also cite Michel et al. 2024 who highlight wetlands and Nisbet et al. 2023a who has a detailed discussion of the wetland hypothesis.

**Response**: We agree and have cited accordingly in these lines (see above).

Line 78 – solar backscatter: somewhere in the paper or supplemental information there should be a comment on the constant thick cloud cover over the wetter parts of the Congo basin during the rainy seasons, making the boundary layer invisible from remote sensing at times of highest wetness (and presumably productivity). It is very difficult to see the Congo and Amazon wetlands when they are most busy. The Sudd, Okavango and Niger inland delta are exceptions as they are river-fed, far from the rainfall.

**Response**: We agree that this is an important point to make, and in lines 366-369 we now discuss this difference in observation density with reference to Figure 2b.

Line 90 – August peak of emissions? That depends on latitude zone. While it's likely true for the NH north of about 10N (but including the Sudd) it is not true for most Southern hemisphere wetlands (Okavango excepted), nor for parts of the equatorial zone (2 wet seasons, 2 dry seasons annually). During the missing data episodes, was scaling to seasonality done by retrieval latitude? Or just one-size-fits-all?

**Response**: We thank the reviewer for pointing out this issue. In line 91-92 we now clarify that northern hemisphere methane is sharply rising during August due to wetland emissions. In lines 106-108 we clarify that our scaling procedure is done by gridcell, so it accounts for different seasonalities in different regions and latitude bands

Line 112 – WETCHART - I'm concerned that over the Upper Congo Bangweulu wetland, Shaw et al (2022) found a huge discrepancy between their measured results and model estimates from WETCHART and GCP. Now Bangweulu is papyrus/reed wetland at about 11°South and about 1100m altitude, and the Sudd is papyrus swamp about 5-10°N and a warmer 400m altitude, but they are broadly comparable. Shaw et al said "The models underestimated emissions by a factor of 10 on average when compared with fluxes derived from the airborne measurements". Note that France et al (2022) also got very high emissions. In other words I'm concerned that WETCHART has its problems in modelling tropical wetlands. Shaw et al. suggest "that land surface models currently lack the ability to accurately predict emissions from vegetated tropical wetlands in Africa" Are there any field campaign tests to show LPJ-MERRA performs better?

**Response**: We thank the reviewer for this important point. We mention the Shaw2022 paper now in lines 349-351 in the context of prior emissions errors in Africa. LPJ-MERRA2 does not appear to do much better in the Congo, and may be worse (it generally predicts less methane emissions in Congo basin than WetCHARTs). However, to our knowledge field studies of tropical African wetlands have not been explicitly

compared to this version of LPJ. As the reviewer pointed out above, our methodology is able to correct prior errors in the river-fed wetlands but due to cloud cover may be less likely to correct underestimates in the Congo.

Table 1 and Line 118-127 (Priors) – part of the problem is that in Africa the emissions from livestock can be indistinguishable from wetland emissions – same place, same seasons, same isotopes. In dry areas wetlands are few and cattle eat tree foliage and often stray far from water (especially now as boreholes are common), but in water-rich regions like the Zambian wetland and South Sudan the wetlands are full of both cows and antelope (which are ruminant) (not to mention pseudo-ruminant hippos also, and camels in some areas). In a large wetland like the Sudd, the only difference between wetland methane and cow methane is that the wetland source moves a bit more slowly. Same location and same d13C.

**Response**: The reviewer raises a good point and a fundamental source of uncertainty in source attribution. We add a discussion in lines 127-129 to flag the issue and additional discussion throughout the paper (see below) to better characterize what can and cannot be known about source attribution.

Line 123 and Line 238 – do you really think scaling South Sudan cattle to 0.1° x 0.1° is valid? I have never been to South Sudan but I've flown over the area, and I've been all around (Uganda, N. Sudan, Ethiopia) and talked to people with knowledge – I'd be surprised if anyone has valid estimates of South Sudan cattle populations, especially not in unrest areas. FAO's basically a wild guess. African wetlands are full of cows, but the woodland and scrub around the wetland are also full of browsing cows and goats. Moreover, there are 6 million antelope seasonally migrating (and ruminating) through the Sudd, if you believe the African Parks estimate (https://www.africanparks.org/worlds-largest-land-mammal-migration-confirmed-south-sudan). Maybe that's an overestimate, maybe not. The manuscript doesn't make clear how these ruminant population issues are assessed. Populations of kob and reedbuck are likely rapid responders to good nutrition from water-driven growth. So gridding down is a tough call.

**Response**: The reviewer is right about this uncertainty in inventories, especially in South Sudan. We clarified in lines 270-272 that we do not rely on this sub-grid attribution alone in our discussion, and will instead bring in additional data sources (like GRACE) to better separate drivers.

Line 125 – geological scaling to Hmiel – yes, agree.

**Response**: We are glad to hear this!

Line 129-134 and also line 148 – OH is a key part of the paper but skimmed over – the discussion is all about sources but very little about sinks. I'm especially concerned about the year 2020. While Qu et al 2022 (cited in ms) attributed only 14% of the surge to OH changes, Peng et al. 2022 state they "attribute the methane growth rate anomaly in 2020 relative to 2019 to lower OH sink (53  $\pm$  10 per cent) and higher natural emissions (47  $\pm$  16 per cent), mostly from wetlands." Also note Bouarar et al 2021 – something big was going on with OH and needs to be

considered. Finally, note Morgenstern et al on 14CO. More generally, yes, OH is a huge and ill-constrained topic in its own right, and maybe it's wise to bite off one chunk of the problem at a time (e.g. sources), but these 5 lines do gloss over a whole mega-can of OH worms (and soil sinks, as well as Cl which has big fractionation on 13C).

**Response**: We agree with the reviewer that this is an important source of uncertainty in this paper, and we also agree that OH is a large and difficult topic on its own that may be best approached in separate studies. We moved line 148 and expanded it into a new paragraph (lines 149-155) discussing study limitations arising from our holding OH constant. We also added the citations to Bouarar2021, Morgenstern2025, and Peng2022 in addition to a few other papers to better engage with the existing literature.

Line 130 – OH lifetime – specify the type of lifetime (steady state vs perturbation)

**Response**: We clarified in line 143 that we are talking about the steady-state lifetime.

Line 148 – 'we do not optimize OH'....Line 149 – 'prevent constraining OH as a local variable'? – explain? This may be a key factor in the 2020 surge – see Peng et al and also Bouarar et al.

**Response**: We thank the reviewer for pointing out this point of confusion. In addition to more generally discussing how OH contributes to uncertainty in this study (lines 149-155), we also clarify why we think it is difficult to simultaneously optimize OH and methane emissions with LETKF (lines 149-152), which is due to the localization assumptions of the algorithm.

Line 249 – attributing between livestock and wetland in the Sudd? Maybe if you think the cattle have been shot in the unrest and ongoing military violence there, but it seems an impossible task to distinguish between [cows+goats+antelope] and [wetland]

**Response**: We agree that this is a difficult source attribution problem, so in lines 270-272 and 346-348 we add the caveat that prior inventories are highly uncertain in the region, so our discussion in the results will make use of other data sources.

Line 296-301 – biogenic emissions, not OH. See also Nisbet et al. 2023.

**Response**: We agree that our findings are consistent with a surge in biogenic emissions, even without our post-assimilation analysis (i.e. prior inventory distributions point to biogenic emissions from livestock and waste). We expanded our discussion of African wetlands in what is now lines 323-369 to more clearly suggest that wetlands specifically are important in driving the surge.

Line 301 – livestock growth in 2020s – see Nisbet et al 2025.

**Response**: We add a citation to Nisbet et al., 2025 in lines 361-363 in the context of discussing livestock population growth in eastern Africa.

Line 305 – use of GRACE – good...

Response: We agree.

Line 320-332 – yes, despite my scepticism about the livestock data, this paragraph sounds plausible. But it might be good to add real data about the huge floods in both South Sudan and Sudan (e.g. 2020).

**Response**: We are glad that the reviewer agrees, as the inundation argument in this section does not require a highly accurate livestock prior.

Line 318 – repeat - I really don't think you can split livestock and wetland emissions in the Sudd. Intricately intermixed and interwoven, and I suspect d13C is the same.

**Response**: We add a discussion in lines 346-348 agreeing with the reviewer that source attribution is difficult in this region, but that nonetheless inundation data can offer a clue as to the principal drivers of change over this unique study period.

Line 324 – note that while the water in the Sudd is from the White Nile (and Lake Victoria was pouring out at record highs between April and July 2020), the Blue Nile is also a factor. In 2020 the Blue Nile Flood level reached over 17m at Khartoum. When you fly over the join at Khartoum you can see how the cloudy White Nile's steady capacitor discharges into the drk Blue Nile's dramatic peak flow and very small low-water river flow. Khartoum is at 381m altitude, while Malakal is at 385m and Bor is at 407m. so the Blue Nile flood backs up the White for months until the Ethiopian flood subsides in September...Yes, the Blue Nile has been affected by the enormous GERD dam filling in Ethiopia, but nevertheless in 2020 the Blue Nile would have had quite an impact on the lower northern part of the Sudd wetlands.

**Response**: We thank the reviewer for pointing out this interesting effect. In lines 357-358 we now mention the Blue Nile effect on the rate of White Nile discharge with an appropriate citation.

Line 343 – when it's wet the trees have lots of leaf for browse and the palatable grasses grow tall – livestock eat when it's wet, and when its dry they starve, so the co-seasonality of livestock and wetland is to be expected.

**Response**: The reviewer raises an interesting point here, and we mention the possibility of cattle browsing in phase with wetland emissions in line 381.

Line 343 – not yet much rice in Africa though production is growing fast.

**Response**: The reviewer is correct on this point, and we have cited Chen et al., 2024 to illustrate that this production is increasing (lines 383-384). Our overall finding on rice discussed here is a northern hemisphere mean and thus is more strongly affected by emissions in southern and eastern Asia.

Line 368 – Siberian gas production is (was) seasonal – used to do maintenance in high summer, and then fill up German pipes in the last quarter. See Reshetnikov paper.

**Response**: We were not aware of this pattern, and have cited the Reshetnikov2000 paper as a possible exception to the aseasonality of fossil emissions (lines 408-409).

Line 393 – assuming no OH change....NO, you can't assume that. It clearly changed but we don't know by how much. You can say you left it out of the calculation, and then make some generalised remarks to say if OH went down by say 10% in mid 2020, then the atmospheric methane burden would go up by X Tg. etc etc.

**Response**: We thank the reviewer for bringing this up. We modified this statement in lines 435 to make more clear that we left OH change out of our calculation. This is in addition to our new paragraph on the methane sink in lines 149-155 discussed earlier. We also now discuss the impact of our constant OH assumption on our emissions calculation in lines 298-299.

Line 399 - 403 – maybe here you could mention the failure of models to fit the measurements by Shaw et al, and the need to develop better models!

**Response**: We agree with this observation and added lines 445-447 to make the point.

I found figure 8 a bit disconcerting. Maybe some attention also to Russia?

**Response**: The reviewer raises an interesting point. We don't see much seasonality in Russian grid cells dominated by oil and gas, but this may be due to poor observational capacity in the winter (lines 417-418).

**Some References to consider**

Bouarar, I., et al. 2021. Ozone anomalies in the free troposphere during the COVID-19 pandemic. *Geophysical Research Letters*, 48(16), p.e2021GL094204.

Drinkwater, A., et al. 2023. Atmospheric data support a multi-decadal shift in the global methane budget towards natural tropical emissions, *Atmos. Chem. Phys.*, 23, 8429–8452, https://doi.org/10.5194/acp-23-8429-2023.

France, J.L., et al. 2022. Very large fluxes of methane measured above Bolivian seasonal wetlands. *Proc Nat Acad Sci USA*, 119(32), p.e2206345119.

Lunt, M.F., et al. 2021. Rain-fed pulses of methane from East Africa during 2018–2019 contributed to atmospheric growth rate. *Environ Res Lett*, 16, p.024021.

Michel, S.E., et al. 2024. Rapid shift in methane carbon isotopes suggests microbial emissions drove record high atmospheric methane growth in 2020–2022. *Proceedings of the National Academy of Sciences*, 121(44), p.e2411212121.

Morgenstern, O., Moss, R., Manning, M., Zeng, G., Schaefer, H., Usoskin, I., Turnbull, J., Brailsford, G., Nichol, S. and Bromley, T., 2025. Radiocarbon monoxide indicates increasing atmospheric oxidizing capacity. *Nature Communications*, 16(1), p.249.

Nisbet, E.G., et al. 2023a. Atmospheric methane: Comparison between methane's record in 2006–2022 and during glacial terminations. *Global Biogeochemical Cycles*, *37*(8), e2023GB007875.

Nisbet, E.G., 2023b. Climate feedback on methane from wetlands. *Nature climate change*, *13*, 421-422.

Nisbet, E.G. 2025. Practical paths towards quantifying and mitigating agricultural methane emissions. *Royal Soc Proc A*, 481, p. 20240390).

Palmer, P.I., et al. 2023. Drivers and impacts of Eastern African rainfall variability. *Nature Reviews Earth & Environment*, 4, 254-270.

Peng, S., et al. 2022. Wetland emission and atmospheric sink changes explain methane growth in 2020. *Nature*, 612, pp.477-482.

Reshetnikov, A.I., Paramonova, N.N. and Shashkov, A.A., 2000. An evaluation of historical methane emissions from the Soviet gas industry. *J Geophys Res: Atmos*, 105, 3517-3529. Shaw, J.T., et al. 2022. Large methane emission fluxes observed from tropical wetlands in Zambia. *Global Biogeochemical Cycles*, 36(6), p.e2021GB007261.

Zhang, Z., et al 2024. Ensemble estimates of global wetland methane emissions over 2000–2020. *Biogeosciences*, 22, 305–321, https://doi.org/10.5194/bg-22-305-2025.

**Response**: We thank the reviewer for these references. We have cited all of them at appropriate places in the revised manuscript.

**Reviewer 2**

Pendergrass and co-authors present an analysis of the global methane budget for years 2019-2023 that is derived from satellite observations and an ensemble-based source inversion methodology. They examine patterns of seasonal and interannual variability, and draw conclusions regarding particular sources contributing to the inferred emissions. The methodology is novel, the paper is well-written, and the analysis contributes to an ongoing community assessment of recent methane source and sink changes. The topic is suitable for ACP. I feel the issues below should be addressed prior to publication.

**Response**: We thank the reviewer for their support of our manuscript, and for their constructive criticism. We have responded to the issues raised by the reviewer below. All line numbers refer to the clean revised manuscript without tracked changes.

I appreciate the concision of the paper but feel that the role of uncertainty is given short shrift. The set of inversions includes runs with two different wetland flux estimates and runs optimizing emissions + concentrations versus emissions alone. The authors do discuss some caveats in their results and point out unrealistic patterns that likely arise from uncertain prior emission distributions. But overall I feel the paper needs a more thorough and systematic assessment of uncertainties in the results. Do the authors feel that the inversion suite covers the true uncertainty space (I doubt it)? How can we assess the robustness of the derived spatial, temporal, and sectoral conclusions?

Response: The reviewer raises an important issue. We agree that that our inversion ensemble does not capture all aspects of the uncertainty in our results. In the revision, we have highlighted some important sources of uncertainty that are difficult to quantify but should inform interpretation of our results. In lines 149-155 we now discuss uncertainty arising from our constant OH sink. Because of strong OH error correlations we are not confident in the ability of LETKF to simultaneously constrain OH and methane emissions, but this introduces uncertainty due to unaccounted OH changes (also now discussed in lines 297-299 and 435). We also now highlight the difficulty of source separation particularly in eastern Africa (lines 270-272, 346-348) and more clearly frame our discussions of additional data sources (e.g. GRACE-FO) as a way of performing source attribution despite uncertainty. By highlighting these two especially important sources of uncertainty, and discussing what interpretations are favored by other data sources, we believe the revised manuscript is clearer on the robustness of our results.

One of the main points of this paper relates to the interannual variability of methane emissions from 2019-2023. There is approximately a month of missing TROPOMI data during July and August in both 2022 and 2023; the authors point out that this is the time of year when methane emission is at its peak. Equation 1 is used as a correction factor to account for this issue when computing annual fluxes and IAV, but this approach assumes that the peak emission season does not contribute significantly to IAV in the annual flux. This seems like a questionable assumption and is certainly a caveat that should be acknowledged. Is there a way to use other datasets to test the viability of this assumption? For example, do the seasonal concentration anomalies in July-August from GOSAT or NOAA look similar in 2021 to those in 2022-2023?

**Response**: We thank the reviewer for raising this point. We looked at the monthly mean for the NOAA marine surface background sites and found that the 2021 July/August mean had a similar relationship to the annual 2021 mean as in 2022 and 2023, with some variability. This discussion is now in lines 98-102 with appropriate citation.

OH fields driving methane loss are scaled to match the 11.2 year MCF lifetime. Does this mean that the same OH fields are used for each year, or that OH is simulated separately for each year, and the output for each year is then scaled uniformly to match the MCF constraint? I guess the difference between these two would not be very big (just in the OH spatial distribution) but please clarify for the reader.

**Response**: We thank the reviewer for pointing out this source of confusion. We clarified that indeed we use the same sink throughout the inversion (i.e. without interannual variability) in lines 145-146.

In either case, the inversion period spans the COVID era when there were large pollutant emission changes (e.g., NOx, etc.) that affect OH. So employing OH fields that do not vary between years does seem justified here. What is the rationale and potential impact of this?

**Response**: We thank the reviewer for pointing out the issue. In response, we have expanded our rationale of excluding OH from the optimization into a paragraph in lines 149-155. In short, excluding OH adds to the uncertainty of our results, but we are not

confident that OH and methane can be simultaneously optimized due to regional error correlation. We discuss the impact of this exclusion on the interpretation of our results in a few places in the revised manuscript, such as lines 297-299 and 435.

"Figure 7c shows that northern tropical emissions seasonality corresponds with the cycle of mean GRACE-FO water storage data". Based on Figure 7c this is somewhat debatable. For example, emissions increase steadily from January through April whereas inundation is decreasing during that time. Inundation does increase thereafter. But, since the slope of the emissions trend is more or less stable throughout January-September during which time the inundation has periods of both decrease and increase, one might instead conclude that the emissions are unaffected by whatever is happening to inundation.

**Response**: We thank the reviewer for bringing up this point, and we certainly agree that other drivers besides inundation of wetlands will affect methane emissions. We narrowed our claim in the interpretation of figure 7c to suggest that the emissions and inundation seasonal peaks coincide in lines 378-379.

Is an additional LETKF burn-in period required following a ~month of missing data as in the cases above? It would help to have a bit more explanation of how these data gaps were handled in the assimilation.

**Response**: The reviewer raises an important point. We edited the paper in lines 103-106 to say that the LETKF rapidly adjusts to correct emissions (i.e. within two 5-day assimilation windows) after the period of missing data, eliminating the need for an extended burn-in. This may because the system is already well-adjusted to observations by 2022/2023, and, as noted in the text, our run-in-place approach re-uses observations and reruns parts of the observation window, which gives the system time to adjust after the observations return.

Line 112: "the nine-member high-performance subset of the WetCHARTs v1.3.1 inventory ensemble", I assume this means you are using the mean of this ensemble? Please clarify.

**Response**: Yes, we use the mean. We clarify this in line 122 of the revised manuscript.

Line 136 and 160, "state vector of emissions and/or concentrations". As I understand it, the state vector either contains emissions, or emissions + concentrations. The "and/or" implies that you also use a state vector with just concentrations, which I don't think is the case. So the wording can be clearer.

**Response**: In both places raised by the reviewer, we replaced "emissions and/or concentrations" with "emissions, or concatenated emissions and concentrations" to clarify the wording.

Line 215, I think that negative emissions are not intrinsically unphysical for methane (e.g., soil uptake), though presumably the magnitude of such an effect would not be detectable from space. Consider more precise wording.

**Response**: The reviewer raises a good point that methane fluxes could be negative. We adjusted our wording in lines 234-235 to clarify that in our setup negative emissions for methane are unphysical given that we do not optimize our soil sink.

---

## Author Response (AR2)

Thank you for revising the manuscript. The reviewers have the following suggestions before acceptance of the manuscript:

**Response**: We have responded to the remaining issues raised by the reviewers below. All line numbers refer to the clean revised manuscript without tracked changes.

-The uncertainties could be handled more comprehensive. The reviewer recommend the authors to add a paragraph to the conclusion section summarizing "study limitations and remaining uncertainties".

**Response**: We thank the reviewer for this suggestion. We have added a paragraph summarizing limitations and remining uncertainties to the conclusion in lines 453-461.

-Chen et al. (2025) and references therein show a reduction in OH in 2020 due to COVID-19. Please discuss further how this influence your results.

**Reference:**

Chen, W., Zhang, Y., & Liang, R. (2025). Converging evidence for reduced global atmospheric oxidation in 2020. National Science Review, nwaf232.

**Response**: We thank the reviewer for calling attention to this important factor and for introducing us to this recent paper, which we have cited. We have added a discussion on how a reduction of OH in 2020 would impact our results both in the body of the article (lines 298-300) and in the conclusion (lines 453-461).